# Hydrogen decreases susceptibility to AngII-induced atrial fibrillation and atrial fibrosis via the NOX4/ROS/NLRP3 and TGF-β1/Smad2/3 signaling pathways

**Binmei Zhang**[1], **Jingxiu Hou**[2], **Jiaren Liu**[3], **Junhui He**[1], **Yunan Gao**[1], **Guangnan Li**[1], **Tianjiao Ma**[4], **Xin Lv**[5], **Li Dong**[1]*, **Wei Yang**[1]*

1 Department of Cardiology, The Fourth Affiliated Hospital of Harbin Medical University, Harbin, Heilongjiang, China, 2 Department of Cardiology, The First Affiliated Hospital of Chongqing Medical University, Chongqing, China, 3 Department of Clinical Lab, The Fourth Affiliated Hospital of Harbin Medical University, Harbin, Heilongjiang, China, 4 Department of Cardiology, Nangang Branch of Heilongjiang Provincial Hospital, Harbin, Heilongjiang, China, 5 Department of Cardiology, The First Affiliated Hospital of Harbin Medical University, Harbin, Heilongjiang, China

* dongli127@163.com (LD); hydyangwei@hrbmu.edu.cn (WY)

**Data Availability Statement:** All relevant data are within the paper and its Supporting Information.

**Funding:** This work was supported by the research projects of Fourth Affiliated Hospital of Harbin

## Abstract

Atrial fibrillation (AF) represents the commonly occurring cardiac arrhythmia and the main factor leading to stroke and heart failure. Hydrogen ($H_2$) is a gaseous signaling molecule that has the effects of anti-inflammation and antioxidation. Our study provides evidence that hydrogen decreases susceptibility to AngII-mediated AF together with atrial fibrosis. Following continuous AngII administration for a 28-day period, AngII+$H_2$ treated rats showed decreased susceptibility to AF, a decrease in atrial fibrosis, a decrease in ROS in atrial myocytes, an inhibition of NLRP3 inflammasome activation, an improvement in electrical remodeling, and an inhibition of proliferation and migration of cardiac fibroblasts. We further found that hydrogen regulates the activation of inflammasome and thus improves $Ca^{2+}$ handling and IKAch and IKur by inhibiting the activity of NOX4 in vivo. In addition, hydrogen was involved in AngII-mediated atrial fibrosis through inhibiting TGF-β1/Smad2/3 pathway through suppressing TGF-β1 activation and secretion in vivo. Our findings suggest that hydrogen is important for preventing and treating AngII-mediated AF and atrial fibrosis, suggesting that hydrogen could be used as the candidate way to prevent and treat AF.

## Introduction

Atrial fibrillation (AF) represents the commonly occurring cardiac arrhythmia, while prolonged atrial fibrillation can induce stroke or even heart failure [1]. AF occurs via complex mechanisms, including oxidative stress, inflammation, and fibrosis [2]. Notably, AF occurrence is strongly affected by renin-angiotensin-aldosterone system (RAAS) [3]. Angiotensin (AngII) is a major RAAS effector, which activates ROS generation, TGF-β1/smad2/3 pathway, as well as the NLRP3 inflammasome via NADPH oxidase, which in turn promotes ion channel

Medical University [grant number HYDSYTB202125]. The funders had no role in study design, data collection and analysis, decision to publish, or preparation of the manuscript.

**Competing interests:** The authors have declared that no competing interests exist.

dysfunction, abnormal calcium handling and fibrosis. These changes induce AF genesis and progression [4]. Hence, NADPH oxidase is a central mediator of atrial remodeling and atrial fibrillation. NADPH oxidase 4 (NOX4) affects electrical and structural AF remodeling by promoting the generation of ROS [5].

The NACHT, LRR, and PYD domain-containing protein 3 (NLRP3)-inflammasome accounts for a key inflammatory signal transduction complex regulating the intrinsic immune response [6]. NLRP3 inflammasome promotes the abnormal release of $Ca^{2+}$ from the sarcoplasmic reticulum (SR) by upregulating p-CaMKII and CaMKII-dependent p-RyR2 [7]. NLRP3 also promotes ion channel dysfunction, including IKur (Altra-rapid delayed rectifier $K^+$ current) together with IKAch (Acetylcholine-activated inward-rectifier $K^+$ current) [8]. AngII-mediated TGF-β1 up-regulation within fibroblasts plays an essential role in AngII-mediated cardiac fibrosis [9]. Ang-II-mediated oxidative stress (OS) also enhances TGF-β1/Smad2/3 pathway activity [10].

Hydrogen is an important bioregulator that displays effects like antioxidation, anti-inflammation and anti-apoptosis. Hydrogen protects against disorders like cancer, ischemia-reperfusion injury, atherosclerosis, heart failure, together with diabetes [11]. However, studies on the effect of hydrogen on cardiac arrhythmias are limited [12].

Our findings showed that hydrogen considerably decreased AngII-induced AF and the prevalence and course of AF. Hydrogen probably inhibits NOX4 expression and ROS production, thus preventing NLRP3 inflammasome activation while inhibiting electrical remodeling. Additionally, hydrogen reduces TGF-β1 production and activity, suppresses TGF-β1/smad2/3 pathway and prevents structural remodeling. Overall, our results suggested that hydrogen might be used for treating AF.

## Materials and methods

### Experimental animals

The animal experiments approval was done by the Ethical Committee on Animal Experimentation of Harbin Medical University, Harbin, China, under the approval No. 20200915. All the procedures were carried out as per the guidelines stipulated by the Helsinki Declaration (1975) and the National Science Council of the Republic of China. Animal welfare was observed throughout this study by ensuring humane handling and well-being of the animals were incorporated into the design and conduct of all procedures during the period of this research [13]. We obtained Sprague Dawley (SD) male rats (weight, 180–200 g) at the Animal Experimentation Center of Harbin Medical University (Harbin, China) and raised them at 24˚C and under the light/dark cycle of 12-h/12-h. All rats had free access to standard laboratory chow and drinking water.

### Implantation of an osmotic mini-pump, inhalation of H2

All animals were randomly placed in one of four groups (n = 10 each), which included control, $H_2$, AngII or AngII+$H_2$ group. AngII and AngII+$H_2$ groups were given AngII (1,080 μg/kg/24 h) through an implanted osmotic pump (model 2ML, Alzet, USA) for 28 days. Saline-loaded osmotic pumps were implanted into the rats in the control and $H_2$ groups [14].

As reported previously, 2% hydrogen demonstrated safety and effectiveness, thus, we used that concentration of $H_2$ in this study [15]. Specifically, 2% hydrogen was obtained by mixing air pump-generated air with hydrogen generator-generated hydrogen, followed by administration into boxes containing $H_2$ and AngII+$H_2$ treated rats. The boxes that contained control and AngII rats were filled with air. Rats in all boxes were treated for 6 h every day for 28 consecutive days.

## Echocardiography and blood pressure measurements

After 28 days, rats were anesthetized with pentobarbital (1%, 50 mg/kg, intraperitoneally) for echocardiography and blood pressure recording. The anesthetized rats were sacrificed intravenously with potassium chloride (75 mg/kg) for in vivo experiments. We used a high-resolution EPIQ5 ultrasound system containing an S12–4 (4–12 MHz) probe (Philips Healthcare Ultrasound, Netherlands) for Doppler echocardiography in the 2D and 3D modes for evaluating cardiac function and diameter. The parameters recorded during echocardiography included left atrial area (LA area), left atrial diameter (LAD), together with ejection fraction (EF) of all rats. In this study, BL-420 Biological Functional Experimental System (ChengDu TECHMAN Software, China) was used to measure and record diastolic and systolic blood pressures (DBP and SBP) in the tail artery.

## Electrophysiological study

We performed electrophysiological examinations on animals following the methods described in another study [16]. After the rats were anesthetized, we inserted the electrode catheter (1.9F, Scisense) in the right atrium, and then, provided electrical stimuli with the 100-ms pacing cycle length [17]. We wanted artificially induced AF condition to include fast atrial arrhythmias, accompanied by an irregular ventricular rhythm lasting more than 60 s. AF induction rate along with duration was measured. AF susceptibility was referred to as 10 times of induction rate.

## Cell culture

HL-1 atrial cardiomyocytes (BNCC (BNCC288890 and rat fibroblasts (BNCC354039) were cultured with Dulbecco's modified eagle medium (DMEM) in 10% fetal bovine serum (FBS) at 37°C in 5% $CO_2$. The HL-1 cells were then seeded in six-well plates and treated under various conditions after obtaining 70–80% confluence. HL-1 cells in control group were cultured in DMEM for 24h, those of $H_2$ group were grown in DMEM for 24 h under 75% hydrogen, while cells in the AngII group were treated with AngII (1.0μM) for 24h. In addition, the AngII + $H_2$ group were treated with AngII (1.0μM) under 75% hydrogen for 24 h. Cells of AngII +GLX351322 (inhibitor of NOX4) group were subjected to 24-h treatment using AngII (1.0μM) and GLX351322 (10μM), while those of Ang+GLX351322+$H_2$ group were treated using AngII (1.0μM) and GLX351322 (10μM) under 75% hydrogen for 24 h. Rat fibroblasts of control group were cultivated within DMEM for a 24-h period, those of $H_2$ group were subjected to 24-h culture within DMEM under 75% hydrogen, those of AngII group were subjected to 24-h treatment using AngII (1.0μM), while those of AngII+$H_2$ group were subjected to 24-h treatment using AngII (1.0μM) under 75% hydrogen.

## Intracellular ROS measurement

Intracellular ROS levels were detected using 2′,7′-Dichlorofluorescin diacetate (DCFH-DA, CA1410, Solarbio, Beijing, China) fluorescent dye, as reported elsewhere [18]. In summary, DCFH-DA is a non-fluorescent compound that diffuses freely through the cell membrane and can be hydrolyzed to DCFH by intracellular esterase. The DCFH (non-fluorescent compound) could be oxidized by intracellular ROS to a fluorescent DCF. Thus, the intracellular ROS quantity is correlated with the fluorescent DCF intensity. Following the incubation for 24 h with $H_2O_2$, the cells were rinsed gently and thrice with serum-free medium. Later, DCFH-DA, (final concentration of 10 μM) in medium without serum was added to the HL-1 cells and incubated in the dark for 30 min at 37°C. Finally, the cells were washed in serum-free medium

thrice to eliminate unloaded DCFH-DA probe, and later incubated with DAPI for 10 min. the cells were then imaged using a laser confocal microscope (Zeiss LSM780, Germany).

## Detection of intracellular $Ca^{2+}$ content

Intracellular Ca2+ content detection was done as described elsewhere [19]. The HL-1 cells were cultured in DMEM media with 10% FBS. Cell culture media was then removed and approximately $1x10^5$ /ml of cells were washed with PBS. Later, 2 μM of Fluo-3 (S1056, beyotime) diluted in serum-free media was added to the cells for 1.5 h at 37˚C, 5% CO2. The Fluo-3 was later removed, and the cells were washed twice with PBS. Finally, the cells were resuspended in the culture media and observed under fluorescence microscopy (magnification, x20) to detect [Ca2+]i.

## Evaluation of fibroblasts migration and proliferation

**Transwell migration assay of fibroblasts.**   Fibroblast migration was evaluated as described elsewhere [20]. In summary, 100 μl ($1×10^5$ cells/ml) of cells diluted in serum-free DMEM was seeded into the upper chamber of the Transwells. Later, 500 μl DMEM supplemented with 10% FBS was added into the lower chambers. Next, the Transwell chambers were cultured for 36 h in 5% $CO_2$ at 37˚C. Afterward, chambers were removed, and cells on the upper layer of the chamber were removed using a cotton swab. The chambers were washed twice in PBS and fixed in 4% paraformaldehyde for 30 min at room temperature. The cells were then stained in 1% crystal violet for 10 min at room temperature, inverted on glass slides, and dried. The number of cells that passed through the membrane was determined under an optical microscope (magnification, ×200). The cells were counted in 4 randomly selected high-power fields. The experiment was done in triplicates to obtain the mean ± standard deviation.

**Fibroblast proliferation assay.**   Fibroblast proliferation assay was done as described elsewhere [21]. Briefly, the cytotoxicity of $H_2$, AngII, AngII+ $H_2$, AngII+GLX351322+$H_2$, or AngII +GLX351322 was determined using MTT colorimetric assay (Dojindo). HL-1 cells ($1.0 × 10^4$ cells/well) were seeded in 96-well plates. After culturing HL-1 cells for 24 h, the cells were treated under the following conditions; $H_2$, AngII, AngII+ $H_2$, AngII+GLX351322+$H_2$, or AngII+GLX351322 and then added to the growth medium. After incubation of 24h and 72h, the old medium was replaced with a fresh medium containing 5 mg/mL of MTT and the cells were incubated for 3 h. Next, the cells were washed two times with phosphate-buffered saline (PBS) to remove detached dead cells and the formazan crystals were solubilized in DMSO. Absorbance was measured using a microplate reader at 570 nm. The relative cell viability (%) compared to non-treated cells was calculated by [abs] sample/[abs] control × 100.

**Elisa assays.**   To detect oxidative stress in atrial tissue, commercial kits (Nanjing Jiancheng BioEngineering Institute, China) for MDA (an oxidant-dependent lipid peroxidation marker) and SOD (an antioxidant enzyme) were adopted for ELISA assays according to the instructions. To measure active TGF-β1 levels within fibroblast culture supernatants, rat TGF-β1 ELISA Kit (SEKR-0012, Solarbio) was used following the manufacturer's protocol.

The rat TGF-β1 ELISA procedure involved preparing samples and reagents, including the wash buffer, standards, and detection antibody. Latent TGF-β1 were activated by acidifying the samples and then neutralizing them according to the kit instructions. Samples were diluted with the Sample Diluent (1:100) in order to produce values within the dynamic range of the assay. Standards and samples were added to a microplate coated with a capture antibody specific for TGF-β1. After incubation, the plate was washed to remove unbound substances. A detection antibody was then added and incubated, followed by another wash step. Streptavidin-HRP was introduced, bound to the detection antibody, and the plate was washed again. A

substrate solution was added, leading to a color change due to the enzymatic reaction. The reaction was stopped with a stop solution, and the absorbance was measured using a microplate reader. The concentration of TGF-β1 in the samples was determined by comparing their absorbance values to a standard curve generated from known concentrations of TGF-β1 standards [22].

GEN5 CHS 3.03 software was used to analyze ELISA data by importing absorbance readings from a microplate reader and defining the plate layout with designated standards, controls, and samples. It generated a standard curve using known TGF-β1 concentrations. The software then calculated sample concentrations by comparing their absorbance to the standard curve, and adjusting for dilution factors. Quality control checks ensured the curve's accuracy and validated control results. The software exported the analyzed data and generated reports of ELISA data processing for accurate and efficient quantification of TGF-β1.

**Histological analysis.**    After 24-h fixation within the 4% paraformaldehyde (PFA), atrial tissues were embedded in paraffin and sliced into sections (5 μm thick). Then, sections were stained using Sirius Red staining as well as Masson's Trichrome following standard procedures [23]. Dihydroethidine (DHE) staining used for myo was used to stain cryosections at 37˚C for 30 min. (KM0102, BaiAoLaiBo Technology Company, China) [23]. Atrial sections were later subjected to immunohistochemistry analysis with anti-NOX4 (14347-1-AP, Proteintech, 1:200), anti-NLRP3 (ab214185, Abcam, 1:200), anti-collagen I (bs-0578R, Bioss, 1:200), anti-collagen III (bs-0549R, Bioss, 1:200) [24], and anti-a-SMA (bs-10196R, Bioss, 1:200) [23]. Immunofluorescence staining for NLRP3 (ab214185, Abcam) was performed with HL-1 cells and for a-SMA (bs-10196R, Bioss) with fibroblasts as described previously [25].

**Western blotting (WB) assay.**    The proteins collected in rat atrial tissues and fibroblasts were treated with RIPA lysis buffer that contained protease/phosphatase inhibitors, followed by quantification with a BCA protein concentration kit. Then, 80 μg of the extracted total protein was subjected to SDS-PAGE for separation, followed by transfer onto the PVDF membrane. Thereafter, primary antibodies, which included NOX4 (14347-1-AP, Proteintech, 1:1000), p22phox (sc-271968, Santa Cruz Biotechnology, 1:200), NLRP3 (ab214185, Abcam, 1:1000), ASC (sc-514414, Santa Cruz Biotechnology, 1:200), caspase1-p20 (22915–1-AP, Santa Cruz Biotechnology, 1:1,000), phospho-CaMKII (ab182674, Abcam, 1:1000), phospho-RyR2 (Ser2814,GTX00626, Genetex, 1:1000), Kv1.5 (YT2507, ImmunoWay, 1:1000), Kir3.1 (YT2475, ImmunoWay, 1:1000), Kir3.4 (YT2475, ImmunoWay, 1:1000), TGF-β1 (ab215715, Abcam, 1:1000), phospho-TGFβR I (YP1191, ImmunoWay; 1:1000), phospho-TGFβR II (A1415, Abclonal, 1:1000), phospho-Smad2 (YP1185, ImmunoWay; 1:1000), phospho-Smad3 (YP0585, ImmunoWay; 1:1000), α-SMA (bs-10196R, Bioss, 1:1000), collagen I (bs-0578R, Bioss, 1:500), collagen III (bs-0549R,Bioss, 1:500), and GAPDH (10494-1-AP, Proteintech, 1:10000), were added for overnight membrane incubation under 4˚C in the shaker. Finally, membranes were washed by TBS-T and scanned with an imaging lab system. The Image J software was used for quantifying the results.

**Quantitative real-time PCR (qRT-PCR).**    Total atrial tissue RNA was extracted using Trizol (Seven, Beijing, China), which was then prepared into the first-strand cDNA with All-in-one First Strand cDNA Synthesis Kit Ⅲ for qPCR (Seven, Beijing, China) following the manufacturer's protocol. Thereafter, Kv1.5, Kir3.1, Kir3.4, collagen I, collagen III mRNA expression was measured by qPCR analysis using Two Step RT&qPCR Kit (Seven, Beijing, China) on a 7500 Real-Time PCR System (QuantStudio™ Design & Analysis Software, USA). The cycling conditions for qPCR were as follows: 95˚C for 30 sec, 40 cycles at 95˚ C for 10 sec followed by 60˚˚C for 34 sec. The melting curve was drawn within the temperature range of 60-95˚C. GAPDH was used as an internal control. S1 Table displays sequences of all primers (Sangon

Biotech, Shanghai, China) utilized in this work. mRNA levels were quantified by relative quantification and analyzed using the 2-ΔΔCq method.

## Statistical analysis

The data were presented as mean ±SD. Data among several groups were compared by one-way ANOVA and post hoc student-Newman-Keuls tests (GraphPad Prism 7.0), while data from two groups were compared by unpaired Student's t-tests. $P < 0.05$ was regarded as statistically significant.

## Results

### Hydrogen reduced AngII-mediated susceptibility to AF and AF duration, improved AngII-induced atrial structures in rats

No spontaneous AF occurred in the rats of any group. The results of the atrial electrophysiological examinations showed that the AF susceptibility and AF duration of AngII + $H_2$ rats decreased relative to AngII rats, (Fig 1A–1C). Then, we found that after 28-day injection of AngII, LA area of AngII rats enlarged, while hydrogen inhalation was slightly reduced, with no significant difference. LAD of AngII + $H_2$ group decreased relative to AngII group, (Fig 1D and 1F). However, the EF did not show any significant difference among rats in four groups (Fig 1G). It was also found that the SBP decreased in AngII + $H_2$ group compared to AngII group; regardless of the slightly lower DBP, the difference did not exhibit any significance, (Fig 1H and 1I). Therefore, hydrogen treatment decreased AngII-induced AF and showed a protective effect on the atrial size.

### Hydrogen-inhibited AngII-induced atrial oxidative stress in rats

The MDA concentration in the atrial tissue of AngII + $H_2$ group considerably decreased relative to AngII group. In contrast, SOD activity was higher in the rats of the AngII + $H_2$ group compared with AngII group (Fig 2A and 2B). Immunohistochemistry examinations and Western blotting assays showed that NOX4 levels decreased in AngII + $H_2$ group relative to AngII group (Fig 2C–2F). Also, the expression of the p22phox (a membrane subunit that promotes the activation of NOX4 [27, 28]) decreased in AngII + $H_2$ rats compared to AngII group rats, as determined by Western blotting assays. Additionally, ROS production (as demonstrated by DHE staining) of AngII rats was significantly increased in the AngII compared to AngII + $H_2$ rats (Fig 2G). These results indicated that hydrogen acted as an antioxidant in the AngII-induced rats and decreased ROS production.

### Hydrogen suppresses NLRP3 inflammasome activation in rats

We first determined that NLRP3 expression increased in AngII rats compared to Con and $H_2$ groups by performing IHC of NLRP3, compared to a significantly lower NLRP3 of AngII+$H_2$ group (Fig 3A). The ASC, NLRP3, mature IL-1β and Caspase1-p20 levels within AngII rats increased relative to Con and $H_2$ groups, and the levels were lower in the rats of the AngII + $H_2$ group relative to those in the rats of the AngII group. (Fig 3B–3F) Thus, hydrogen inhibited NLRP3 inflammasome activation by inhibiting ROS production.

### Hydrogen improves AngII-mediated atrial electrical remodeling

Activation of NLRP3 inflammasome promotes abnormal $Ca^{2+}$ handling and electrical remodeling [13]. Based on WB analysis, CaMKII phosphorylation of AngII group considerably increased (due to NLRP3 activation) compared to that in the rats of the AngII + $H_2$ groups. A

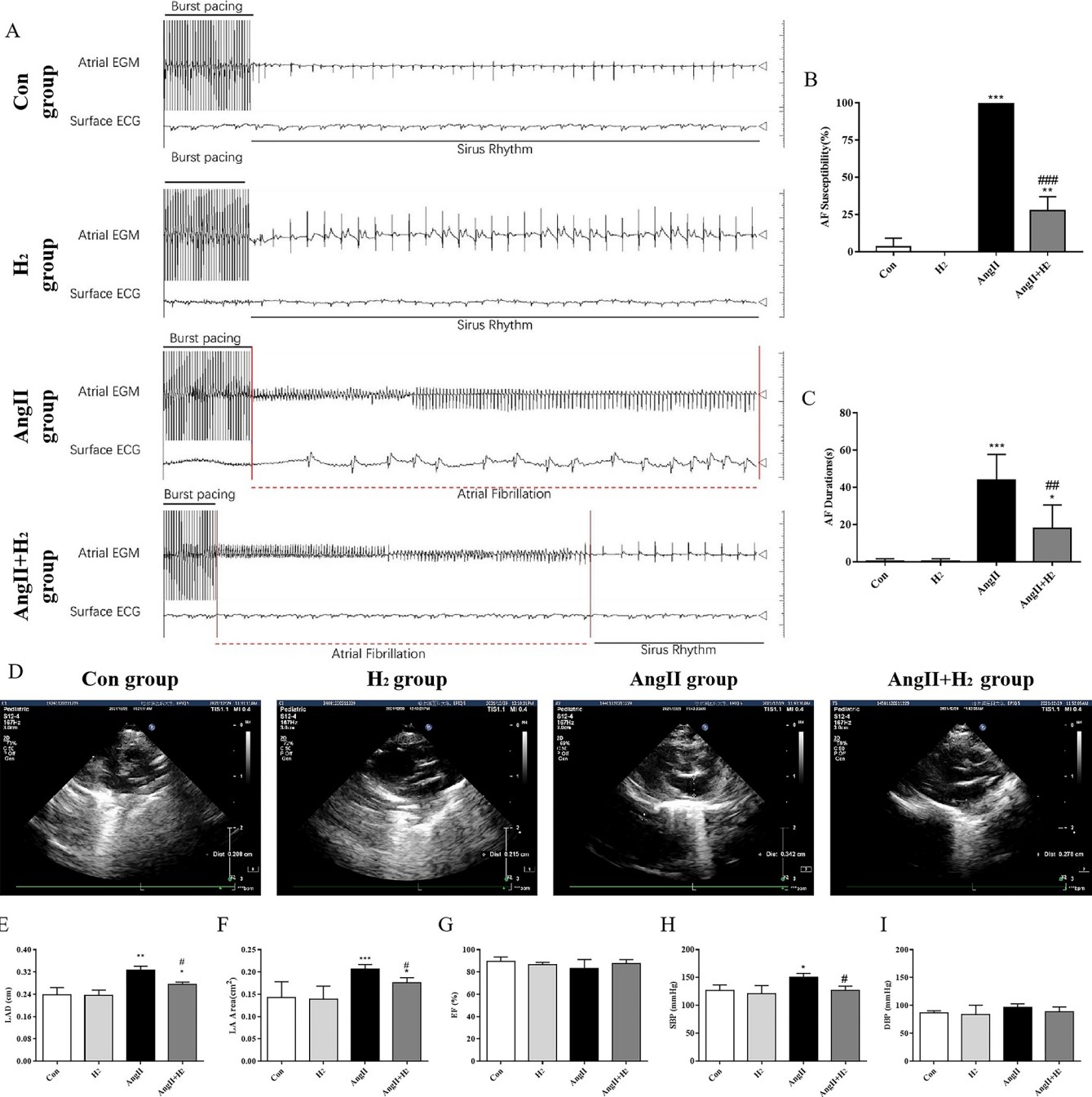

**Fig 1. Hydrogen reduced AngII-induced susceptibility and duration of atrial fibrillation and improved atrial structure and blood pressure in AngII-induced rats.** (A) An atrial electrogram and a surface electrocardiogram. The high solid line represents Burst pacing. Atrial fibrillation is denoted as the red dotted line, whereas sinus rhythm is denoted as the black solid line. (B) and (C) Quantification of AF susceptibility and AF duration of the rats in the four groups. (D) The ultrasound examination of the left atrium of the rats in the four groups. (E) and (F) The left atrial area and internal diameter. (G) The ejection fraction of the heart. (H) and (I) The SBP and DBP in the tail artery of the rats in the four groups; (n = 5). *$P < 0.05$, **$P < 0.01$, ***$P< 0.001$, versus Con group; #$P < 0.05$, ##$P < 0.01$, ###$P < 0.001$, versus AngII group.

similar pattern was found for the phosphorylation of RyR2 associated with p-CaMKII, which regulates the release of $Ca^{2+}$ in the sarcoplasmic reticulum (Fig 4A–4C). We also examined Kir3.1, Kir3.4 ($I_{K-Ach,}$), and Kv1.5 ($I_{Kur}$) protein levels that regulated by the aberrant expression

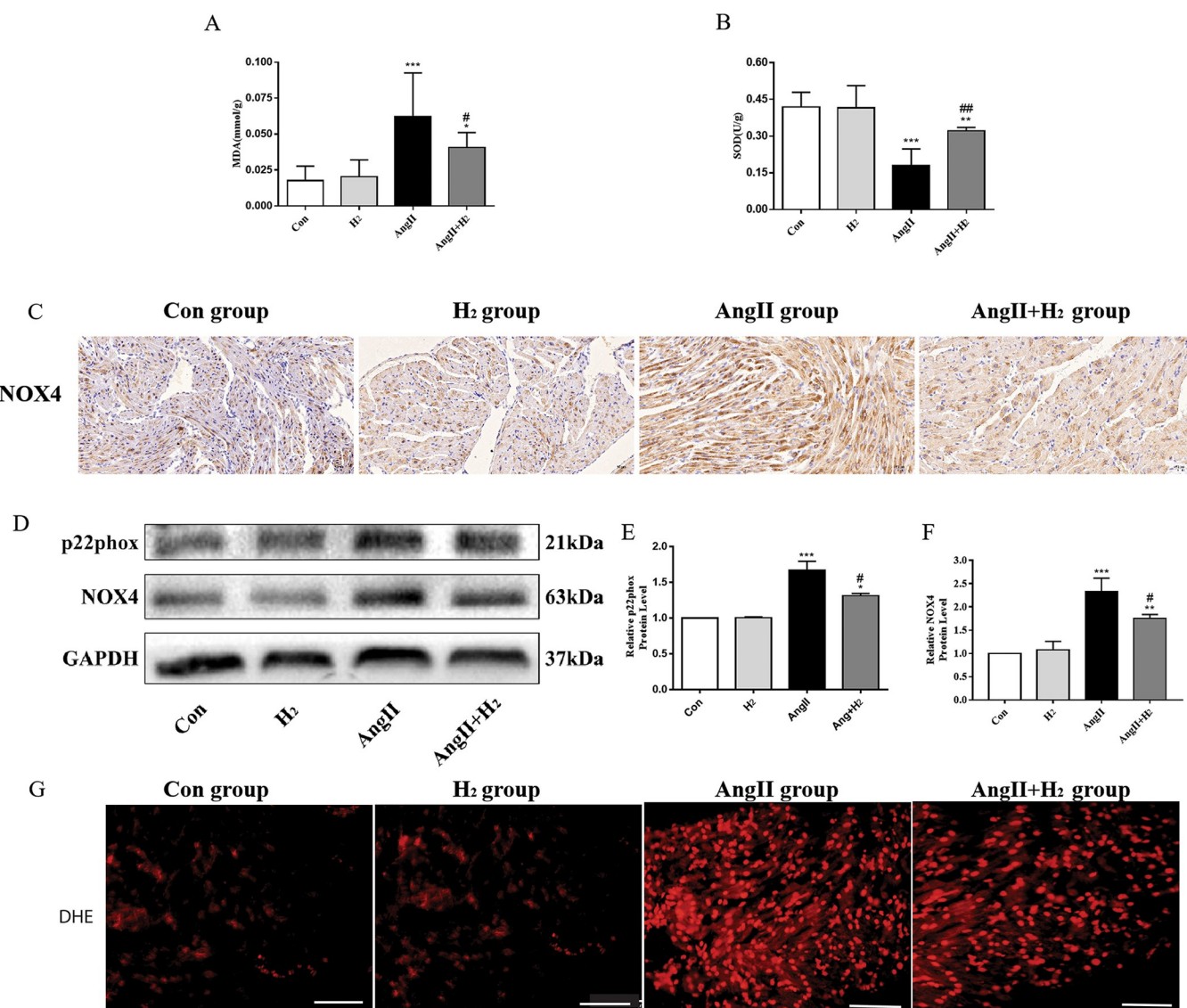

**Fig 2. Hydrogen inhibited oxidative stress in AngII induced rats.** (A) The MDA levels and (B) the SOD activities in the atrial tissue, as determined by the ELISA assay. (C) NOX4 (brown) expression in the atria, determined by immunohistochemical staining. (D) The Western blotting bands showing NOX4 and p22phox (E). (F) Relative NOX4 and p22phox protein level. (G) DHE staining showing ROS expression (red) in the atrial tissue; (n = 3). $*P < 0.05$, $**P < 0.01$, $***P< 0.001$, versus Con group; $\#P < 0.05$, $\#\#P < 0.01$, $\#\#\#P < 0.001$, versus AngII group.

of NLRP3 [13, 29–31]. We also found that the expression of the above-mentioned potassium ion channel in AngII + $H_2$ group decreased relative to AngII group. (Fig 4D–4G) These results showed that treatment with hydrogen enhanced the release of AngII-induced abnormal $Ca^{2+}$ and electrical remodeling via ROS-activated NLRP3 inflammasome.

## Hydrogen reduces AngII-mediated atrial fibrosis in rats by TGF-β1/smad2/3 pathway

After continuously injecting AngII for 28 days, left atrial fibrosis area of AngII group increased relative to AngII + $H_2$ group, determined by Masson staining (Fig 5A). According to Sirius Red staining, the collagen level in the left atrium of AngII group considerably increased

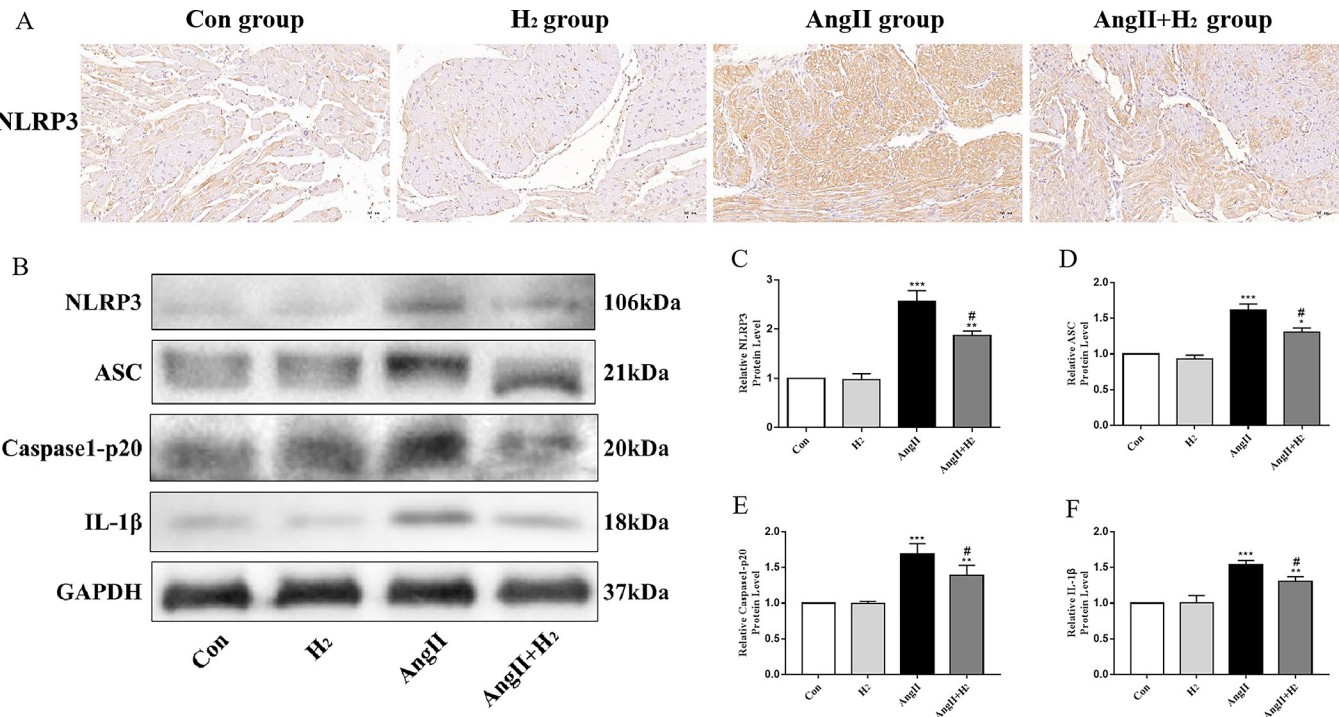

**Fig 3. Hydrogen suppresses NLRP3 inflammasome activity in AngII-mediated rats.** (A) NLRP3 (brown) levels in the rat atria in each group, determined by immunohistochemical staining (×200). (B) Western-blotting bands for NLRP3, ASC, Caspase1-p20 and mature IL-1β. (C-F) The relative levels of ASC, NLRP3, Caspase1-p20 and mature IL-1β proteins; (n = 3). *$P < 0.05$, **$P < 0.01$, ***$P< 0.001$, versus Con group; #$P < 0.05$, ##$P < 0.01$, ###$P < 0.001$, versus AngII group.

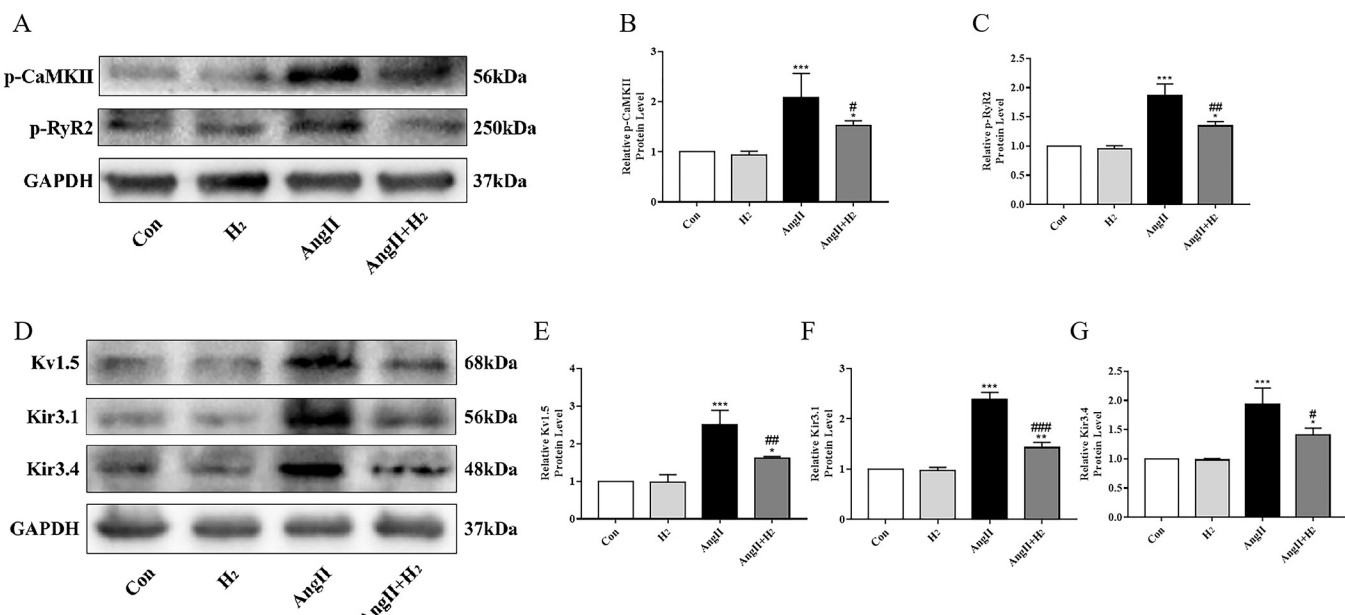

**Fig 4. Hydrogen improves atrial electrical remodeling in AngII-induced rats.** (A) Western blot bands showing p-CaMKII together with p-RyR2. (B) and (C) The relative p-CaMKII and p-RyR2 protein expression. (D) Western blot bands of Kv1.5, Kir3.1 and Kir3.4. (E-G) The relative levels of Kv1.5, Kir3.1 and Kir3.4 proteins; (n = 3). *$P < 0.05$, **$P < 0.01$, ***$P< 0.001$ versus Con group; #$P < 0.05$, ##$P < 0.01$, ###$P < 0.001$, versus AngII group.

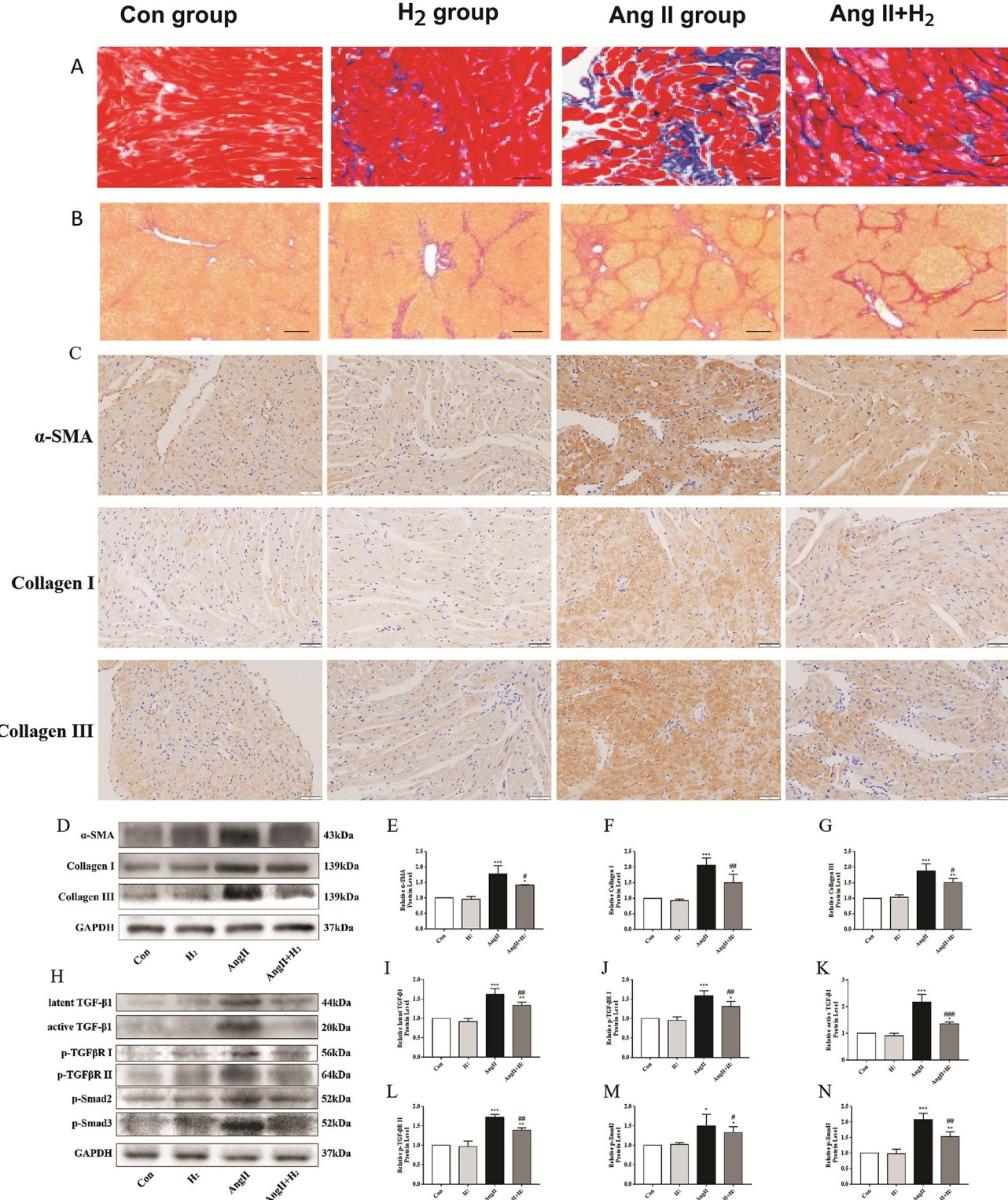

**Fig 5. *Hydrogen reduces AngII-mediated atrial fibrosis in AngII-treated rats* via TGF-β1/smad2/3 pathway.** (A) and (B) Masson and Sirius Red staining showed fibrotic disorders (blue) and collagen deposition (red) in the atria. (C) α-SMA, collagen I, together with collagen III (brown) levels through immunohistochemical staining in atria. (D) Western blotting bands showing α-SMA, collagen I, as well as collagen III. (E-G) Relative α-SMA, collagen I, and collagen III protein expression. (H) Western blot bands and (I-N) relative levels (n = 3) of latent TGF-β1, active TGF-β1, p-Smad2, p-Smad3, p-TGFβR I, and p-TGFβR II. $^{*}P < 0.05$, $^{**}P < 0.01$, $^{***}P < 0.001$, versus Con group; $^{\#}P < 0.05$, $^{\#\#}P < 0.01$, $^{\#\#\#}P < 0.001$, versus AngII group.

relative to AngII + H$_2$ group (Fig 5B). The immunohistochemical examination and WB assay were similar, and the collagen I, α-SMA, and collagen III expression of AngII + H$_2$ group considerably decreased relative to AngII group. These findings suggested that hydrogen inhalation reduced atrial collagen deposition (Fig 5C–5G). TGF-β1/Smad2/3 pathway represents the classical pathway promoting fibrosis and can be secreted and activated by AngII stimulation [15]. The results of the WB assay showed that atrial tissue proteins in the rats of the AngII + H$_2$ group considerably decreased latent TGF-β1, active TGF-β1, p-Smad2, p-Smad3, p-TGFβR I, and p-TGFβR II levels compared with AngII group, and levels of those proteins in the rats of both groups increased relative to Con and H$_2$ groups (Fig 5H–5N). Based on these findings, hydrogen downregulated atrial fibrosis via TGF-β1/smad2/3 pathway.

## Hydrogen ameliorates AngII-mediated electrical remodeling of HL-1 cells via NOX4/ROS/NLRP3 pathway

To verify the role of NOX4 in the protection of hydrogen, we used GLX351322 to inhibit NOX4 and observe the expression of downstream ROS and NLRP3, as well as the effect of electrical remodeling. Relative to control group, both ROS and NLRP3 levels were increased following AngII treatment. However, ROS and NLRP3 decreased in the H$_2$ and GLX351322 treated groups, and the reduction of ROS and NLRP3 was more significant in the combination of H$_2$ and GLX351322 treated groups, (Fig 6A and 6B).

In addition, we examined the intracellular Ca$^{2+}$ concentration and mRNA expression levels of Kv1.5, Kir3.1 and Kir3.4 within HL-1 cells to verify the function of H$_2$ in electrical remodeling. Relative to control group, intracellular Ca$^{2+}$ concentration of AngII group apparently elevated, and mRNA of Kv1.5, Kir3.1 and Kir3.4 was apparently elevated. However, AngII+H$_2$ and AngII+ GLX351322 groups showed decreased Ca$^{2+}$ concentration and decreased mRNA of Kv1.5, Kir3.1 and Kir3.4. In addition, Ca$^{2+}$ concentration and mRNA of Kv1.5, Kir3.1 and Kir3.4 of the AngII+H$_2$+ GLX351322 group also decreased, which did not exhibit better protective effects, compared with the AngII+H$_2$ or AngII+ GLX351322 groups, (Fig 6C–6F).

## Hydrogen suppresses fibroblast differentiation into myofibroblasts via TGF-β1/smad2/3 pathway

The MTT and Transwell assays confirmed that the proliferation and migration rate was reduced in the AngII+H$_2$ compared to the AngII fibroblast group, (Fig 7A–7C). In addition, unexpectedly, α-SMA level of AngII+H$_2$ group was drastically reduced, while collagen I and collagen III mRNA expression were downregulated compared to AngII group, (Fig 7D–7F). We next examined active TGF-β1 expression within fibroblast supernatants and TGF-β1/Smad2/3 pathway-associated protein levels in fibroblasts. Relative to control group, latent TGF-β1 and active TGF-β1 dramatically elevated after AngII treatment for 24 hours, but the expression of both was decreased after the administration of H$_2$ treatment. p-TGFβR I and TGFRβ II, receptors for TGF-β1, had evidently decreased levels of AngII+H$_2$ group relative to AngII group. Their downstream p-Smad2 and p-Smad3 had evidently decreased levels in the AngII+H$_2$ group compared with the AngII group. Interestingly, p-Smad3 expression markedly increased relative to p-Smad2 after AngII treatment of fibroblasts (Fig 7G–7N). Thus, hydrogen inhibits TGF secretion and activation and decreases AngII-induced atrial fibrosis through suppressing TGF-β1/Smad2/3 pathway.

## Discussion

According to our results, hydrogen administration could suppress the occurrence of AngII-mediated AF and atrial fibrosis. Hydrogen treatment reduced the expression of NOX4, which

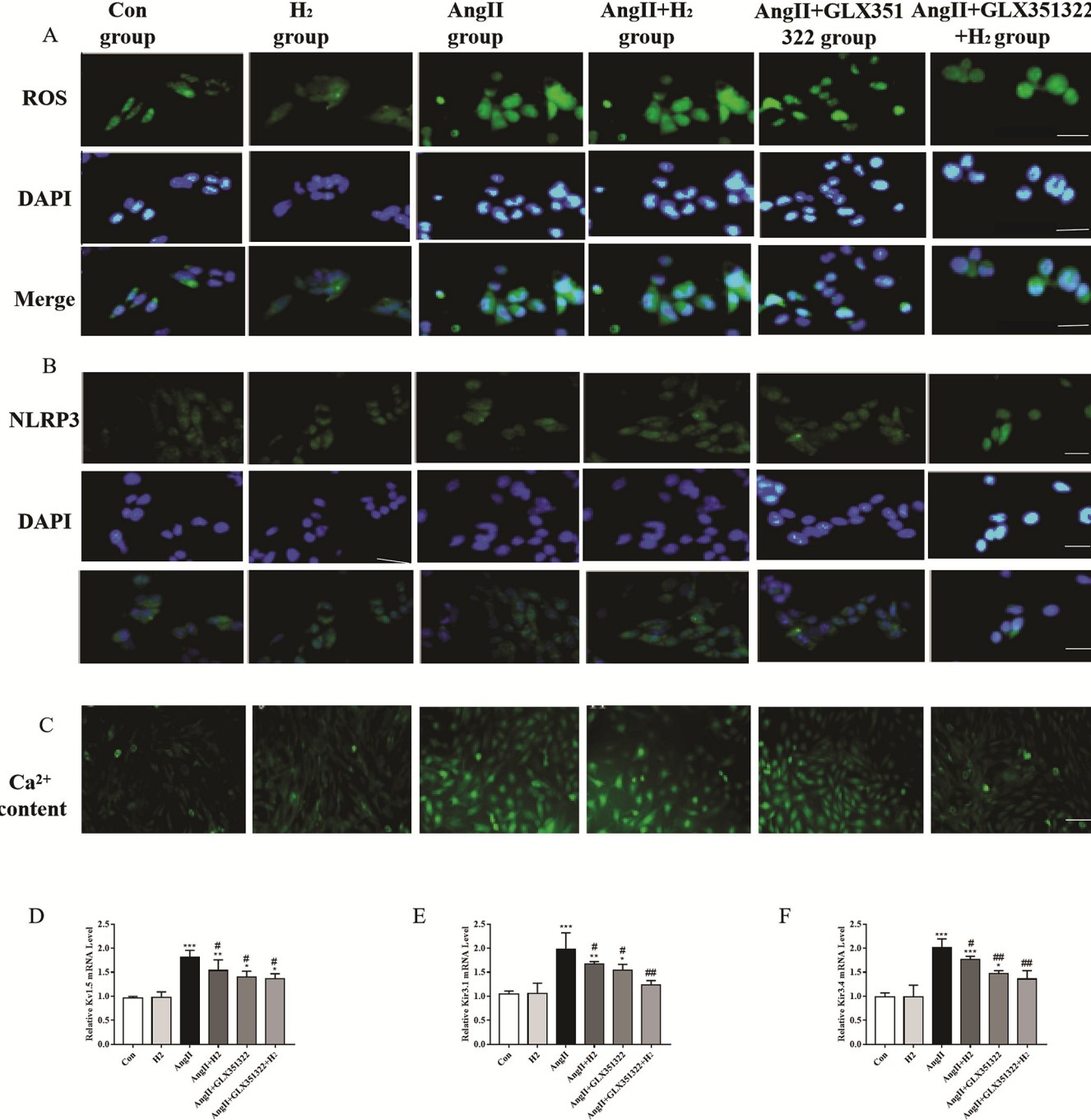

**Fig 6. Hydrogen ameliorates AngII-induced electrical remodeling of HL-1 cells via NOX4/ROS/NLRP3 pathway.** (A) ROS production in HL-1 cells by immunofluorescence staining (×200). (B) Expression of NLRP3 within HL-1 cells by immunofluorescence staining (×200). (C) Intracellular $Ca^{2+}$ concentration in HL cells by fluorescent probes (×200). (D), (E) and (F) mRNAs of Kv1.5, Kir3.1 and Kir3.4 by qRT-PCR. (n = 3). $^{*}P < 0.05$, $^{**}P < 0.01$, $^{***}P < 0.001$, versus Con group; $^{#}P < 0.05$, $^{##}P < 0.01$, $^{###}P < 0.001$, versus AngII group.

increases ROS production, and thus, it suppressed NLRP3 inflammasome. In addition, administration of hydrogen suppressed the secretion and activation of TGF-β1 and inhibited TGF-β1/Smad2/3 pathway, and consequently improved electrical and structural remodeling.

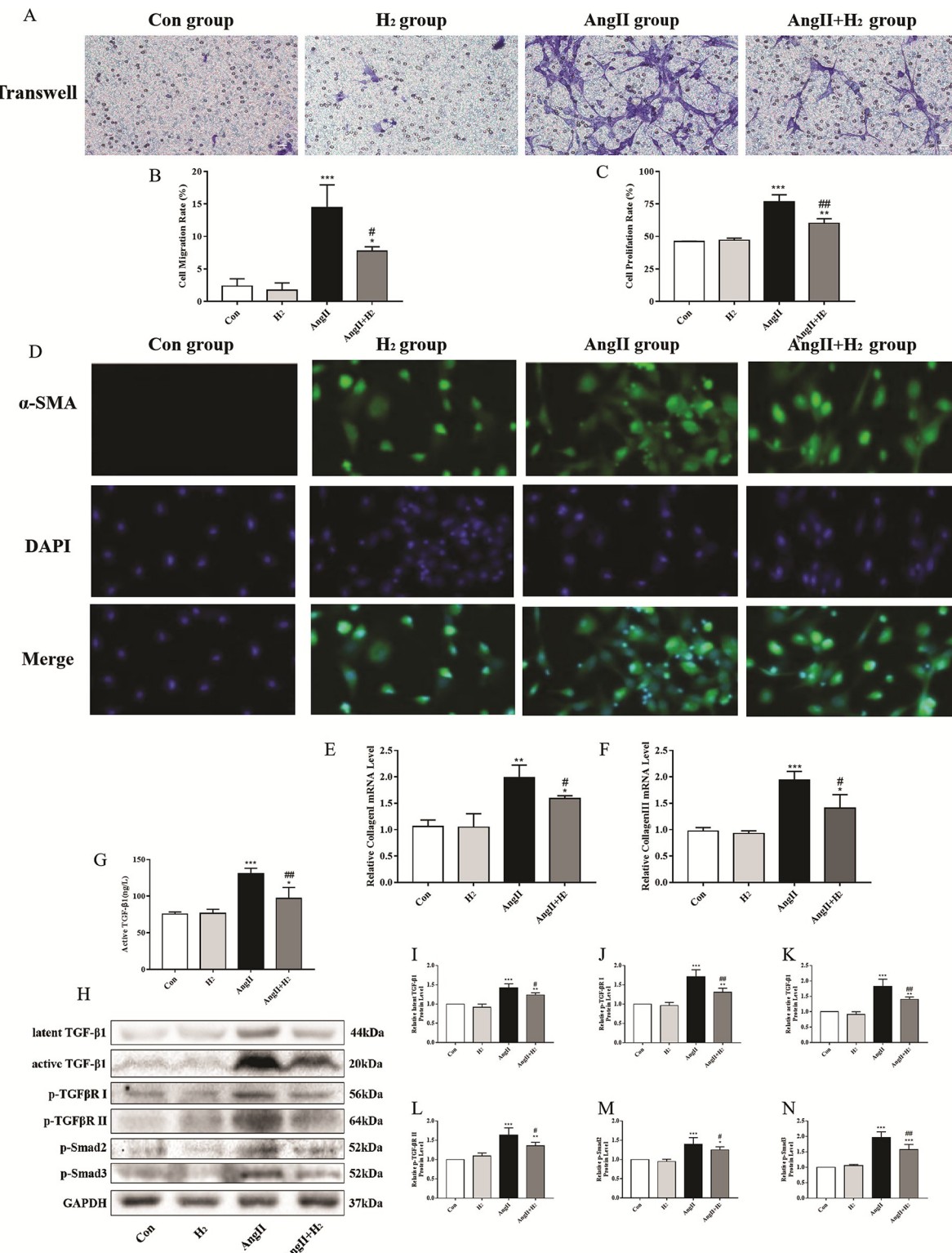

**Fig 7. Hydrogen inhibits fibroblast differentiation into myofibroblasts via TGF-β1/Smad2/3 pathway.** (A) Representative images showing crystalline violet staining via Transwell (×200). (B) Fibroblast migration of atrial fibroblasts; (C) Atrial fibroblast proliferation measured through MTT assay. (D) α-SMA level of each group of fibroblasts by immunofluorescence staining (×200). (E) and (F) Collagen I and Collagen III mRNA expression measured through qRT-PCR. (G) ELISA was conducted to measure active TGF-β1 within fibroblast supernatants. (H) The Western blot bands and (I-N) relative protein levels (n = 3) of latent TGF-β1, active TGF-β1, p-TGFβR I, p-TGFβR II,

p-Smad2 and p-Smad3; (n = 3) $^*P < 0.05$, $^{**}P < 0.01$, $^{***}P < 0.001$, versus Con group; $^\#P < 0.05$, $^{\#\#}P < 0.01$, $^{\#\#\#}P < 0.001$, versus AngII group.

## Hydrogen decreases oxidative stress in AF by inhibiting NOX4 activity

Atrial fibrillation is associated with oxidative stress. Excess ROS levels might be associated with atrial electrical/structural remodeling, which can cause AF and atrial fibrosis [26]. Atrial remodeling indicates AF genesis and development, and it also induces an increase in the LA and abnormal conduction [2]. The process involves multiple mechanisms, such as oxidative stress, inflammation, and atrial fibrosis [27]. Oxidative stress also facilitates focal electrical activity and reentry [28]. AngII can dramatically elevate SBP and increase the risks of dysfunction and left ventricular hypertrophy, and these have been identified as risk factors for AF [4].

NADPH oxidase accounts for the main ROS source during atrial fibrillation, especially NOX2 and NOX4. AngII has an important effect on the activation of NADPH oxidase, and it induces the generation of excessive ROS within cardiomyocytes [29]. NOX4 is expressed in fibroblasts, cardiomyocytes, vascular smooth muscle cells, and endothelial cells. The NOX4 activation only requires binding of the p22phox subunit [30]. Increased NOX4 can significantly upregulate NOX4-derived $H_2O_2$ levels in AF patients. NOX4 affects both structural remodeling and atrial electrical remodeling. It upregulates Kv1.5 by mediating ROS generation, which selectively participates in repolarization in the atria of humans [31]. Luo et al. showed that by increasing ROS and ox-CaMKII, NOX4 could cause abnormalities in calcium signaling, eventually causing atrial fibrillation in rats and mice [6]. Our results showed that hydrogen prevented oxidation in AngII-induced rats. Changes in MDA, ROS, NOX4, and p22phox in the atria suggested that inhalation of hydrogen reduced the damage of AngII to the atria by suppressing OS.

## Hydrogen improved electrical remodelling in atrial fibrillation through suppressing NOX4/ROS/NLRP3 pathway

Electrical remodeling of AF is associated with NLRP3 inflammasome activation [32]. Triggers and substrates are necessary for initiating and maintaining AF, including early along with delayed afterdepolarizations (EADs and DADs); both provide the substrate for AF [33]. The NLRP3 inflammasome can regulate $Ca^{2+}$ handling abnormalities, which depends on DADs. Cardiomyocyte-specific NLRP3 inflammasome activation promotes the aberrant leakage of $Ca^{2+}$ from the sarcoplasmic reticulum (SR) along with electrical remodeling via CaMKII phosphorylation and CaMKII-dependent RyR2 Ser2814 phosphorylation [8]. The aberrant release of $Ca^{2+}$ from the SR provides the substrate for focal ectopic activity in atrial fibrillation [33]. Additionally, NLRP3 inflammasome activation affects ion channels dysfunction in atrial myocytes.

According to Yao et al., activation of the NLRP3 inflammasome can increase IKur and $IK_{Ach}$ production, which are inward rectifying potassium currents, and can shorten action potential duration (APD), decrease effective refractory period (ERP), and facilitate reentry, eventually increasing the incidence of atrial fibrillation [34]. Scott Jr et al. Also came to the same findings with an increase in Kv1.5 (Kur's molecular basis), Kir3.1, and Kir3.4 (two subunits of KAch) levels [35]. Thus, our results suggested that in vitro and in vivo, hydrogen, through its effects of antioxidation and anti-inflammation, suppressed NLRP3 inflammasome activation as well as CaMKII and RyR2 phosphorylation, causing ectopic firing by generating DADs. Treatment with hydrogen also decreased the $K^+$ channel (Kv1.5, Kir3.1, and Kir3.4), which shortened the APD and AERP and created a reentry substrate.

## Hydrogen alleviated structural remodeling and fibrosis in atrial fibrillation through inhibiting TGF-β1/smad2/3 pathway

Angiotensin II can activate and maintain high levels of TGF-β1, an important fibrotic growth factor, in the hearts of fibrotic animals [36]. The target genes of TGF-β1 mainly include collagen I, fibronectin, α-smooth muscle actin (α-SMA, a myofibroblast marker), and connective tissue growth factor. The overexpression of TGF-β1 triggers excessive collagen deposition in the interstitial myocardium, leading to myocardial fibrosis [37].

The TGF-β1 synthesis and its receptor expression are widespread, but the most important part of TGF-β1 signaling is the TGF-β1 activation phase. TGF-β1 activation can be regulated via various conditions, mainly: mechanical stress, the action of specific enzymes, acidic environment, intracellular oxidative stress, and some other factors [38]. According to Yao et al., TGF-β1 activation exacerbated myocardial fibrosis and loss of cardiac function [25]. Canonical TGF-β1 pathway can be regulated by binding active TGF-β1 to phosphorylated TGFβR II. p-TGFβR II is responsible for recruiting and activating TGFβR I phosphorylation, thereby phosphorylating its downstream Smad2/3 [39, 40].

The phosphorylation of Smad2/3, a downstream factor of TGF-β1, induces cardiac fibroblast differentiation in myofibroblasts [41]. Smad2 and Smad3 are two closely related Smad protein family membrane that have important effects on the TGF-β1 pathway. They can be activated by the TGF-β1 receptor and phosphorylated, thereby binding to Smad4 to form a transcriptional complex that regulates gene expression. Although both of them can regulate fibrosis, they have different functions. Smad2 mainly has an essential effect on fibrosis via TGF-β1 signaling pathway, however, Smad3 can also participate in non-TGF-β signaling pathways, such as MAPK and PI3K pathways [42]. In addition, Smad3 can play a role by regulating pro-inflammatory factors and chemokines, thus promoting fibrosis occurrence [43]. Therefore, Smad3 plays a more prominent role in tissue remodeling and fibrosis. This is similar to our results, where we found significantly higher p-smad3 expression than p-smad2 in AngII-induced fibrosis given, but both were correspondingly reduced after $H_2$ administration.

Myocardial fibrosis has an essential effect on atrial remodeling of persistent cases of AF. Atrial fibrosis might also be a cause of focal and reentry arrhythmia [44]. Some studies have shown that hydrogen can improve myocardial fibrosis after myocardial infarction [11]. According to our in vitro and in vivo findings, hydrogen inhibits TGF-β1 secretion and activation, suppresses TGF-β1/Smad2/3 pathway, and ultimately ameliorates myocardial fibrosis.

### Limitations

Our study had some limitations. First, AngII can produce ROS via NOX2 and NOX4 in cardiac myocytes, but this study was conducted only for NOX4 and its subunits. Second, we verified the elevated expression of Kv1.5, Kir3.1, and Kir3.4 in terms of both translation and transcription, but the experimental conditions prevented Mapping and membrane clamp studies to observe the above-mentioned IKur and IKAch current changes. Finally, the current knowledge about the application of $H_2$ is only in animal and cellular experiments and cannot be applied to clinical trials.

### Conclusion

To summarize, according to our results, hydrogen application remarkably ameliorated AngII-mediated AF and atrial fibrosis, which was accompanied by a decrease in NOX4-mediated overproduction of ROS, reduction of the NLRP3 inflammasome, together with

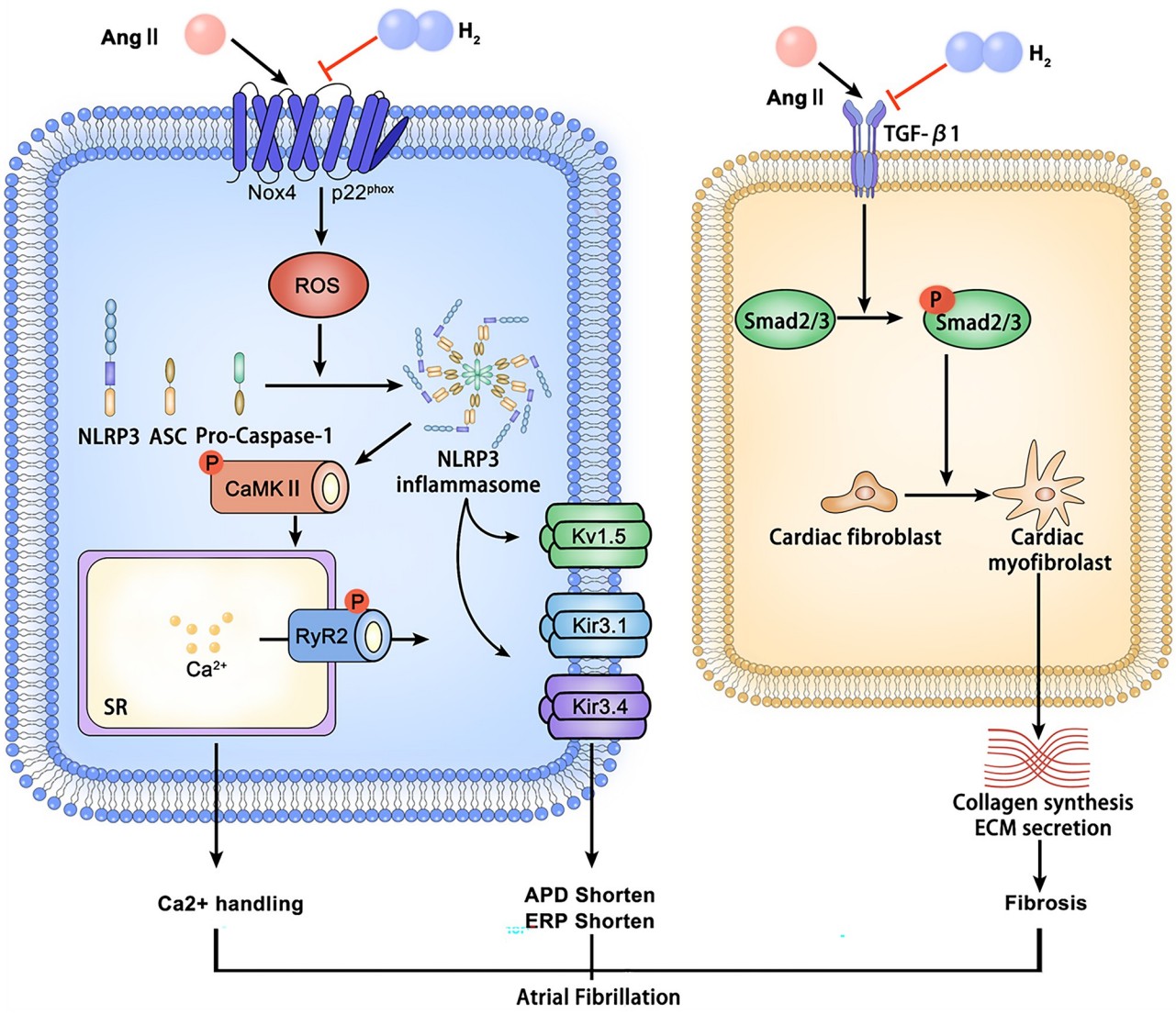

**Fig 8. Mechanism of H₂ inhibition on AngII-mediated AF and atrial fibrosis.**

downregulation of TGF-β1/smad2/3 pathway. These changes improved electrical and structural remodeling, respectively. (Fig 8). Thus, H₂ might be used for treating atrial fibrillation.

## Supporting information

**S1 Table. qRT-PCR primer sequences.**
(DOCX)

**S1 Raw images. Images of the original western blots.**
(DOCX)

## Acknowledgments

The authors wish to thank all the team members in the department of Cardiology, the Fourth Affiliated Hospital of Harbin Medical University for their support.

## Author Contributions

**Conceptualization:** Binmei Zhang, Jiaren Liu, Wei Yang.

**Data curation:** Jingxiu Hou.

**Investigation:** Junhui He, Li Dong.

**Methodology:** Binmei Zhang, Jiaren Liu, Wei Yang.

**Software:** Binmei Zhang, Jiaren Liu, Tianjiao Ma, Xin Lv, Wei Yang.

**Supervision:** Yunan Gao, Guangnan Li.

**Validation:** Tianjiao Ma, Xin Lv.

**Visualization:** Junhui He, Li Dong.

**Writing – original draft:** Jingxiu Hou.

**Writing – review & editing:** Binmei Zhang.

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
