## [Decision Letter · Decision Letter 0]

18 Jun 2024

PONE-D-24-21187Hydrogen Decreases Susceptibility to Ang II-induced Atrial Fibrillation and Atrial Fibrosis via the NOX4/ROS/NLRP3 and TGF-β1/Smad2/3 Signaling PathwaysPLOS ONE

Dear Dr. Yang,

Thank you for submitting your manuscript to PLOS ONE. After careful consideration, we feel that it has merit but does not fully meet PLOS ONE’s publication criteria as it currently stands. Therefore, we invite you to submit a revised version of the manuscript that addresses the points raised during the review process.

We look forward to receiving your revised manuscript.

Kind regards,

Sepiso Kenias Masenga, PhD

Academic Editor

PLOS ONE

Journal Requirements:

"This work was supported by the research projects of Fourth Affiliated Hospital of Harbin Medical University [grant number HYDSYTB202125]. The funders had no role in study design, data collection and analysis, decision to publish, or preparation of the manuscript."

Additional Editor Comments:

As per journal journal guidelines:

1. Relating to the "Financial Disclosure" statement please include Initials of the authors who received each award to clearly show Grant numbers awarded to each author. This statement is required for submission and will appear in the published article if the submission is accepted. Please make sure it is accurate.

2. For Animal Research (involving vertebrate animals, embryos or tissues), Provide the name of the Institutional Animal Care and Use Committee (IACUC) or other relevant ethics board that reviewed the study protocol, and indicate whether they approved this research or granted a formal waiver of ethical approval. If the study involved non-human primates, add additional details about animal welfare and steps taken to ameliorate suffering. If anesthesia, euthanasia, or any kind of animal sacrifice is part of the study, include briefly which substances and/or methods were applied

Human Subject Research (involving human participants and/or tissue). Give the name of the institutional review board or ethics committee that approved the

study. Include the approval number and/or a statement indicating approval of this research. Indicate the form of consent obtained (written/oral) or the reason that consent was not obtained (e.g. the data were analyzed anonymously)

3. We note that your Data Availability Statement is currently as follows: [All relevant data are within the manuscript and its Supporting Information files. [Yes - all data are fully available without restriction]

Reviewers' comments:

Reviewer's Responses to Questions

**Comments to the Author**

1. Is the manuscript technically sound, and do the data support the conclusions?

Reviewer #1: Yes

Reviewer #2: Yes

2. Has the statistical analysis been performed appropriately and rigorously? 

Reviewer #1: Yes

Reviewer #2: Yes

3. Have the authors made all data underlying the findings in their manuscript fully available?

Reviewer #1: Yes

Reviewer #2: Yes

4. Is the manuscript presented in an intelligible fashion and written in standard English?

Reviewer #1: Yes

Reviewer #2: Yes

5. Review Comments to the Author

Reviewer #1: Zhang et al. studied the role of hydrogen in Ang II-induced Atrial Fibrillation and Atrial

Fibrosis and showed that the NOX4/ROS/NLRP3 and TGF-β1/Smad2/3 Signaling Pathways are involved in the pathogenesis. Although they provided strong evidence for hydrogen’s role in improving atrial fibrillation and fibrosis, the data must be improved. I have the following comments and suggestions.

Line 55, please provide the references.

Figure text is not clean; please improve them.

The images in Figure 2 G are unclear, making it difficult to identify any cells. Please enhance the image quality to make the cells more visible.

In Figure 4, quantification is not visible.

Figures 5A and B are unclear; please provide better images where cardiac tissue is recognizable.

All the images in Figure 6 must be improved. The current images are not clear.

Same with Figure 7 D.

Reviewer #2: The study “Hydrogen Decreases Susceptibility to Ang II-induced Atrial Fibrillation and Atrial Fibrosis via the NOX4/ROS/NLRP3 and TGF-β1/Smad2/3 Signaling Pathways” by Zhang et al. investigated the ability of hydrogen in prevention of Atrial Fibrillation and atrial Fibrosis. The study used selective administration of hydrogen and angiotensin II in rats and cell cultures, and employed echocardiography, electrophysiological studies, and biochemical assays to collect pertinent data. Their findings showed that hydrogen reduces susceptibility to Angiotensin II (Ang II)-mediated AF and atrial fibrosis in rats. Rats treated with Ang II and hydrogen showed decreased AF susceptibility, reduced atrial fibrosis, lower reactive oxygen species levels, inhibited NLRP3 inflammasome activation, improved electrical remodeling, and reduced proliferation and migration of cardiac fibroblasts. Hydrogen also enhanced Ca2+ handling and ion channel function by inhibiting NOX4 activity. Furthermore, hydrogen inhibited Ang II-induced atrial fibrosis through the TGF-β1/Smad2/3 pathway. The findings suggest that hydrogen may be an effective treatment for preventing and managing Ang II-mediated AF and atrial fibrosis.

This is a good study, however, to improve its readability and credibility, I suggest the following;

Minor comments

1. The authors should ensure consistent use of abbreviations and terminology, for instance, "AngII" should be consistently written as "Ang II" or "AngII" throughout the manuscript.

2. The authors should work on grammatical errors, and ensure that all abbreviations are in place to improve readability

Major comments

1. The authors should consider specifying the concentrations and passage number of cells used in experiments to ensure consistency and reproducibility of their work since every study is a foundation for the next, clarity is paramount. The following article could offer guidance https://www.ahajournals.org/doi/full/10.1161/01.HYP.27.4.897

2. The protocols for intracellular ROS measurement, intracellular Ca2+ content detection, fibroblast migration, and proliferation assays lack sufficient detail. For reproducibility, the authors should consider including specific details such as concentrations, incubation times, and steps for washing and preparation.

3. The authors should specify the preparation of samples, dilutions used, and steps taken to ensure accuracy and consistency in the ELISA assay. Also, clarify how the GEN5 CHS 3.03 program software was used in the analysis of their findings.

4. The description of the qRT-PCR procedure should include details on the thermal cycling conditions.

5. The statement by the authors that animal care and use protocols were followed is appropriate, however, should consider emphasizing ethical considerations and how animal welfare was maintained throughout the study.

6. PLOS authors have the option to publish the peer review history of their article (what does this mean?). If published, this will include your full peer review and any attached files.

Reviewer #1: No

Reviewer #2: No

---

## [Author Response · Author response to Decision Letter 0]

8 Aug 2024

PONE-D-24-21187

Hydrogen Decreases Susceptibility to Ang II-induced Atrial Fibrillation and Atrial Fibrosis via the NOX4/ROS/NLRP3 and TGF-β1/Smad2/3 Signaling Pathways

PLOS ONE

Dear Dr. Yang,

Thank you for submitting your manuscript to PLOS ONE. After careful consideration, we feel that it has merit but does not fully meet PLOS ONE’s publication criteria as it currently stands. Therefore, we invite you to submit a revised version of the manuscript that addresses the points raised during the review process.

We look forward to receiving your revised manuscript.

Kind regards,

GENERAL RESPONSE

>>>We, the authors of the article entitled ‘Hydrogen decreases susceptibility to AngII-induced Atrial Fibrillation and Atrial Fibrosis via the NOX4/ROS/NLRP3 and TGF-β1/Smad2/3 Signaling Pathways’, received with great appreciation and gladness the reviews made by the experts who took their precious time to positively critique and help us improve our manuscript. We value this process for the role it plays in ensuring a scientifically sound, verifiable and replicable data. Consequently, we have taken our time to go through the manuscript and provided a point-by-point response as raised by the reviewers. The responses provided are as below:

>>> Response

We thank the editor for this guide. We have ensured our manuscript meets the PLOS ONE style requirements. 

>>> Response

We thank the editor for this input. We have stated on the method section that rats were anesthetized with pentobarbital (1%, 50 mg/kg, intraperitoneally). This statement is found in line 84.

"This work was supported by the research projects of Fourth Affiliated Hospital of Harbin Medical University [grant number HYDSYTB202125]. The funders had no role in study design, data collection and analysis, decision to publish, or preparation of the manuscript."

>>>Response

We thank the editor for this guideline. We have included the original uncropped western blot images with this submission, as per the request. During some of our western blot experiments, we determined the expression of more than one protein in a single run/gel, and therefore the membrane was pre-cut according to the molecular weight of the proteins under investigation, due to constraints of resources. We have confidence in our original data, which are verifiable and replicable.

>>>Response

We agree with the editor on this statement. The corresponding author has included an ORCID ID together with this submission. 

>>>Response

We thank the editor for this response. We have included the ethics statement in the methods section only, and not in any other section as recommended. The ethical statement reads as;

‘The animal experiments approval was done by the Ethical Committee on Animal Experimentation of Harbin Medical University, Harbin, China, under the approval No. 20200915. All the procedures were carried out as per the guidelines stipulated by the Helsinki Declaration (1975) and the National Science Council of the Republic of China. Animal welfare was observed throughout this study by ensuring humane handling and well-being of the animals were incorporated into the design and conduct of all procedures during the period of this research.’

Additional Editor Comments:

As per journal journal guidelines:

1. Relating to the "Financial Disclosure" statement please include Initials of the authors who received each award to clearly show Grant numbers awarded to each author. This statement is required for submission and will appear in the published article if the submission is accepted. Please make sure it is accurate.

>>>Response

We thank the editor for this suggestion. We have ensured that the initials of the author who received fundings are captured. As an emphasis, the funding for this research was received by Dr. Yang, the corresponding author. 

2. For Animal Research (involving vertebrate animals, embryos or tissues), Provide the name of the Institutional Animal Care and Use Committee (IACUC) or other relevant ethics board that reviewed the study protocol, and indicate whether they approved this research or granted a formal waiver of ethical approval. If the study involved non-human primates, add additional details about animal welfare and steps taken to ameliorate suffering. If anesthesia, euthanasia, or any kind of animal sacrifice is part of the study, include briefly which substances and/or methods were applied

>>>Response

We are in agreement with the editor on the importance of this instruction. We have included the relevant ethics board that reviewed our study protocol. This statement is found from line 64-69, and it reads as;

The animal experiments approval was done by the Ethical Committee on Animal Experimentation of Harbin Medical University, Harbin, China, under the approval No. 20200915. All the procedures were carried out as per the guidelines stipulated by the Helsinki Declaration (1975) and the National Science Council of the Republic of China. Animal welfare was observed throughout this study by ensuring humane handling and well-being of the animals were incorporated into the design and conduct of all procedures during the period of this research.’

Human Subject Research (involving human participants and/or tissue). Give the name of the institutional review board or ethics committee that approved the

study. Include the approval number and/or a statement indicating approval of this research. Indicate the form of consent obtained (written/oral) or the reason that consent was not obtained (e.g. the data were analyzed anonymously)

>>>Response

We thank the editor for this guideline. Our study did not involve any human subjects. 

3. We note that your Data Availability Statement is currently as follows: [All relevant data are within the manuscript and its Supporting Information files. [Yes - all data are fully available without restriction]

>>>Response

We thank the editor for this guideline. Our submission contains all the raw data used to generate the entire manuscript. 

>>>Response

We thank the editor for this detailed guideline relating to data. We have uploaded the original uncropped western blot data as supplementary data, and we have not used any data owned by a third party. 

Reviewers' comments:

Reviewer's Responses to Questions

Comments to the Author

1. Is the manuscript technically sound, and do the data support the conclusions?

Reviewer #1: Yes

Reviewer #2: Yes

>>>Response

We thank the reviewers for recommending our manuscript as technically sound and for their opinion that our data support the conclusions of this manuscript. We are indeed grateful. 

2. Has the statistical analysis been performed appropriately and rigorously?

Reviewer #1: Yes

Reviewer #2: Yes

>>>Response

We thank the reviewers for acknowledging the appropriateness and rigor of our statistical analysis. 

3. Have the authors made all data underlying the findings in their manuscript fully available?

Reviewer #1: Yes

Reviewer #2: Yes

>>>Response

We thank the reviewers for observing all the data that we have made fully available. 

4. Is the manuscript presented in an intelligible fashion and written in standard English?

Reviewer #1: Yes

Reviewer #2: Yes

>>>Response

We are elated that the reviewers found our manuscript to have been presented in intelligible fashion and standard English. 

5. Review Comments to the Author

Reviewer #1: Zhang et al. studied the role of hydrogen in Ang II-induced Atrial Fibrillation and Atrial

Fibrosis and showed that the NOX4/ROS/NLRP3 and TGF-β1/Smad2/3 Signaling Pathways are involved in the pathogenesis. Although they provided strong evidence for hydrogen’s role in improving atrial fibrillation and fibrosis, the data must be improved. I have the following comments and suggestions.

L

---

## [Decision Letter · Decision Letter 1]

8 Sep 2024

Hydrogen Decreases Susceptibility to Ang II-induced Atrial Fibrillation and Atrial Fibrosis via the NOX4/ROS/NLRP3 and TGF-β1/Smad2/3 Signaling Pathways

PONE-D-24-21187R1

Dear Dr. Yang,

We’re pleased to inform you that your manuscript has been judged scientifically suitable for publication and will be formally accepted for publication once it meets all outstanding technical requirements.

Kind regards,

Sepiso K. Masenga, PhD

Academic Editor

PLOS ONE

Additional Editor Comments (optional):

Reviewers' comments:

Reviewer's Responses to Questions

**Comments to the Author**

1. If the authors have adequately addressed your comments raised in a previous round of review and you feel that this manuscript is now acceptable for publication, you may indicate that here to bypass the “Comments to the Author” section, enter your conflict of interest statement in the “Confidential to Editor” section, and submit your "Accept" recommendation.

Reviewer #1: All comments have been addressed

Reviewer #2: All comments have been addressed

2. Is the manuscript technically sound, and do the data support the conclusions?

Reviewer #1: Yes

Reviewer #2: Yes

3. Has the statistical analysis been performed appropriately and rigorously? 

Reviewer #1: Yes

Reviewer #2: Yes

4. Have the authors made all data underlying the findings in their manuscript fully available?

Reviewer #1: Yes

Reviewer #2: Yes

5. Is the manuscript presented in an intelligible fashion and written in standard English?

Reviewer #1: Yes

Reviewer #2: Yes

6. Review Comments to the Author

Reviewer #1: Zhang et all have addressing my all comments. I do not have any further comments. Thank you. Good luck.

Reviewer #2: (No Response)

7. PLOS authors have the option to publish the peer review history of their article (what does this mean?). If published, this will include your full peer review and any attached files.

Reviewer #1: No

Reviewer #2: No

---

## [Editor Report · Acceptance letter]

6 Nov 2024

PONE-D-24-21187R1 

PLOS ONE

Dear Dr. Yang, 

I'm pleased to inform you that your manuscript has been deemed suitable for publication in PLOS ONE. Congratulations! Your manuscript is now being handed over to our production team.

Kind regards, 

on behalf of

Prof. Sepiso K. Masenga 

Academic Editor

PLOS ONE